# Changing Trends of HIV, Syphilis, and Hepatitis C among Male Migrant Workers in Chongqing, China: Nine Consecutive Cross-Sectional Surveys, 2010–2018

**DOI:** 10.3390/ijerph17030875

**Published:** 2020-01-30

**Authors:** Yujun Wang, Rongrong Lu, Guohui Wu, Rong Lan, Rong Ou, Yangchang Zhang, Mengliang Ye

**Affiliations:** 1Department of Epidemiology and Health Statistics, School of Public Health and Management, Chongqing Medical University, Chongqing 400016, China; cqwangyujun@163.com (Y.W.); lr10769909166@163.com (R.L.); 18702370226@163.com (Y.Z.); 2Institute for AIDS/STD Control and Prevention, Chongqing Center for Disease Control and Prevention, Chongqing 400042, China; lurrong@163.com (R.L.); wgh68803652@163.com (G.W.); 3Library of Chongqing Medical University, Chongqing 400016, China; ourong@cqmu.edu.cn

**Keywords:** male migrant workers, HIV, syphilis, HCV, STIs, Chongqing, cross-sectional survey

## Abstract

Background: Male migrant workers (MMWs) have been reported to be vulnerable to human immunodeficiency virus (HIV) and other sexually transmitted infections (STIs). Chongqing, China is one of the major migration destinations and hotspots of HIV. This study aims to explore the prevalence of HIV, syphilis, and hepatitis C virus (HCV), as well as HIV-related knowledge and behaviors, among MMWs in Chongqing. Methods: Questionnaire surveys were conducted, and blood samples were collected and examined among MMWs selected by two-stage stratified sampling in Chongqing from 2010 to 2018. The Cochran–Armitage trend test was conducted to observe the trends in the prevalence of HIV, syphilis, and HCV, as well as HIV-related knowledge and behaviors. The Chi-square test and Binary Logistic Regression were conducted to observe the distinctions between different groups. Results: The overall HIV prevalence was 0.6% with an increasing trend (0.2% to 0.9%, *p* < 0.001), whereas the overall HCV prevalence was 0.5% with a decreasing trend (0.5% to 0.4%, *p* < 0.001). The overall syphilis prevalence was 1.3% in the ≥50 age group, 1.0% in the 30–49 age group, and higher than 0.3% in the 16–29 group (*X*^2^ = 19.527, *p* < 0.001). An uptrend (80.2%–80.6%, *p* < 0.001) was observed in correct HIV-related knowledge. The 16–29 ((Odds Ratio) OR: 1.575; 95%CI (Confidence Interval): 1.380–1.798; *p* < 0.001) and 30–49 (OR: 1.697; 95%CI: 1.495–1.926; *p* < 0.001) age groups had 1.575 and 1.697 times correct HIV-related knowledge more than the ≥50 age group. The proportion of subjects engaged in commercial sex in the past year (7.7%–13.3%, *p* < 0.001), consistent condom use during this activity (20.5%–54.0%, *p* < 0.001), and condom use in the last commercial sex (48.6%–72.1%, *p* = 0.020) were increasing. The risk of engaging in commercial sex in the past year in the 16–29 age group was 0.768 times (OR: 0.768; 95%CI: 0.643–0.917; *p* = 0.003) less than that in the ≥50 age group. The risk of engaging in non-regular sex in the past year in the 16–29 (OR: 2.819; 95%CI: 2.317–3.431; *p* < 0.001) and 30–49 (OR: 1.432; 95%CI: 1.184–1.733; *p* < 0.001) age groups were 2.819 and 1.432 times more than that in the ≥50 age group. The risk of engaging in anal sex in the past year in the 16–29 age group was 6.333 times (OR: 6.333; 95%CI: 1.468–27.327; *p* < 0.013) more than that in the ≥50 age group. The proportion of consistent condom use during non-regular sex in the past year (10.9%–47.3%, *p* < 0.001) and condom use in the last non-regular sex (40.8%–71.1%, *p* < 0.001) increased remarkably. The possibilities of consistent condom use during commercial sex in the past year in the 16–29 (OR: 2.606; 95%CI: 1.847–3.677; *p* < 0.001) and 30–49 (OR: 1.632; 95%CI: 1.214–2.195; *p* = 0.001) age groups were 2.606 and 1.632 times more than that in the ≥50 age group. The possibilities of condom use in the last commercial sex in the 16–29 (OR: 1.805; 95%CI: 1.258–2.589; *p* = 0.001) and 30–49 (OR: 1.360; 95%CI: 1.016–1.821; *p* = 0.039) age groups were 1.805 and 1.360 times more than that in the ≥50 age group. The possibilities of consistent condom use during non-regular sex in the past year (OR: 1.628; 95%CI: 1.066–2.484; *p* = 0.024) and condom use in the last non-regular sex (OR: 1.671; 95%CI: 1.148–2.433; *p* = 0.007) in the 16–29 age group were 1.628 and 1.671 times more than those in the ≥50 age group, respectively. Conclusion: An upward trend of HIV and a downward trend of HCV were observed among MMWs in Chongqing from 2010 to 2018. We also found an increase in commercial sex and inadequate condom use during high-risk behaviors among this population. The overall syphilis prevalence in the middle-aged and elderly groups was higher than in the young group, and elderly MMWs were more likely to engage in unprotected high-risk behaviors. Thus, targeted STI prevention for MMWs in Chongqing, especially those aged 50 years and above, is urgently needed.

## 1. Introduction

HIV is a major contributor to the global burden of disease. In 2010, HIV was the leading cause of disability-adjusted life years worldwide for people aged 30–44 years and the fifth leading cause for all ages [1]. An HIV/AIDS (acquired immune deficiency syndrome) epidemic has been reported in China, with 627,030 people living with HIV/AIDS and 194,435 deaths up to 30 June 2016 [2]. China is experiencing such an HIV epidemic because it has a large population susceptible to infection by high-risk behaviors such as commercial sex, homosexual anal sex, and non-regular sex [3]. Most previous studies and interventions have focused on commercial sex workers, men who have sex with men (MSM), and intravenous drug users; however, increasing domestic migration is a likely factor contributing to the increasing HIV prevalence in China [4,5,6,7]. “Agricultural” (rural) and “non-agricultural” (urban) are the two household registration categories available in China [8]. Rapid urbanization and industrialization driven by economic growth in China has resulted in large-scale rural-to-urban migration [9,10]. Migrants are individuals of rural origin who migrate to urban areas temporarily in hopes of gaining employment and a chance to reap the benefits of the booming urban economic environment; these migrants are mainly comprised of male migrant workers (MMWs) [11]. Mobility among migrant workers has been found to increase vulnerabilities to sexually transmitted infections (STIs), such as HIV and syphilis [12,13,14,15]. This phenomenon has been explained by the separation from long-term partners and increased contact with high-risk partners, such as commercial sex workers [16,17,18]. Most MMWs migrate to urban areas, where the open cultural environment and availability of convenient services increase their probability of engaging in homosexual anal sex, which partly results in the high infection rates of STIs in this group. The existing household registration system in China hinders migrants from switching from a rural to urban residence permanently. Thus, these migrants become marginalized and cannot access the social welfare available to urban residents, resulting in poor living conditions, adverse employment situations, negligible access to health care services, and lack of social support. These drawbacks increase their risk of contracting HIV [19,20,21]. Nevertheless, the number of migrant workers has been increasing every year, despite the unfavorable working and living conditions.

As HIV, syphilis, and HCV are all sexually transmitted diseases (STDs) with the same high-risk population, the Chinese government has built a comprehensive surveillance system, including the national sentinel surveillance system, case reporting system, and special epidemiologic surveys [22,23,24]. In 2010, the number of national sentinel sites expanded from 600 to 1888, covering all of China’s 31 provinces (autonomous regions and municipalities) and targeting eight groups (drug users, female sex workers (FSWs), men who have sex with men (MSMs), male STI clinic attendees, long-distance truck drivers, antenatal care clinic attendees, young college students, and MMWs) [25]. In 2015, 34,315 MMWs were monitored at 87 national sentinel sites of China, with overall antibody positive rates of HIV, syphilis, and HCV at 0.14%, 0.61%, and 0.41%, respectively [26]. A high HIV prevalence among migrants has been reported in several metropolises of China, such as Shanghai and Zhejiang [4,16]. These cities represent developed areas; however, data on undeveloped southwest China, with low income and educational levels (which can be important predictors of STIs), have been largely ignored. Chongqing is one of the four municipalities under the direct control of the central government of China and serves as the economic and political center of southwest China, which has made Chongqing a magnet for migrant workers [27,28]. The average annual rate for new HIV infections in Chongqing from 2007 to 2012 was 19.7%, which greatly exceeded the national infection rate (3.13%) [29]. To augment the limited studies on MMWs in Chongqing, the present study adopted serial cross-sectional surveys to explore the prevalence of HIV, syphilis, and HCV among this group and investigated their HIV-related knowledge and behaviors from 2010 to 2018. Conducting a detailed exploration of reliable and accurate data is essential, as the findings could be translated into effective public health interventions to combat the STI epidemic in Chongqing.

## 2. Methods

### 2.1. Design

The study was supported by the Center for Disease Control (CDC) in Chongqing, China and approved by the medical ethics committee of Chongqing Medical University (IRB number: 2017016). According to the requirement of the Protocol for National Sentinel Surveillance of HIV/AIDS in China, the sentinel monitoring period is from April to June every year [25]. Each participant was asked to anonymously complete the questionnaire and was provided with a unique study-specific identification number (UID).

### 2.2. Recruitment

We restricted our samples to men, as migrant workers are mainly (60%) male and most (70%) cases of HIV are concentrated in this gender group [8,10,11]. The sample selection was followed by a two-stage stratified sampling strategy. First, we selected the three districts in Chongqing (Yuzhong, Jiangbei, and Jiulongpo) with the largest and densest MMWs among all districts. Second, the sample subjects were recruited from four occupational clusters (construction workers, miners, enterprise workers, and business operators), which together composed a large proportion of Chongqing’s MMWs. The inclusion criteria for the study participants were as follows: (1) aged at least 16 years; (2) household registration not belonging to a local district; and (3) volunteered to participate in this study. Permanent MMWs were excluded, as they had no intention to return to their places of origin and, thus, do not serve as a bridge population for transmission. Before the interviews, the researchers explained to the respondents that the data collected would be confidential and used strictly for academic purposes, and 20 Chinese Yuan was paid as an incentive. To ensure privacy during the survey, one investigator alone interviewed one participant in a private room during their rest time. Informed consent and blood collection for laboratory analysis was obtained prior to the questionnaire survey. The survey took approximately 30 to 75 minutes to complete. Finally, 11,252 MMWs were recruited in our study. 

### 2.3. Measures

The serial cross-sectional surveys comprised the national unified sentinel surveillance questionnaire covering socio-demographics as well as HIV-related knowledge and behaviors; furthermore, sero-testing for HIV, syphilis, and HCV was conducted among the MMWs in Chongqing from 2010 to 2018. 

#### 2.3.1. Socio-Demographics, HIV-Related Knowledge, and Behaviors

The socio-demographic information included age, marital status (never married/married/never married but cohabited/divorced or widowed), household registration (Chongqing/other provinces), occupation (construction worker/miner/enterprise worker/business operator/others), local residence time (<3 months/3–6 months/6–12 months/>1 year), and educational level (primary or lower/junior high school/senior high school or above). 

HIV-related knowledge and behaviors were selected, as HIV was the most serious of the three STDs with common transmission routes in our study. We asked the participants to identify transmission modes from eight questions by answering “Yes,” “No,” or “Unknown”. Correct responses to at least 75% of the questions indicated having adequate correct knowledge. The eight items were as follows: (1) people living with HIV can definitely be identified by observing their physical appearance; (2) mosquito bites can spread AIDS; (3) eating with people living with HIV can lead to an HIV infection; (4) the input of HIV-infected blood can lead to an HIV infection; (5) sharing syringes with people living with HIV can lead to AIDS; (6) children born to HIV-infected women can acquire AIDS; (7) consistent condom use can reduce the risk of HIV transmission; and (8) sexual intercourse with only one partner can reduce the spread of AIDS. The internal consistency of this scale was 0.83.

To understand sexual behavioral patterns, we collected information about the respondents’ experience in commercial sex with a female, non-regular sex with a female, and anal sex with a male in the past year (i.e., the respondents answered with a “yes” or “no”). Non-regular sex was defined as sexual intercourse with another person without pay or plans for a long-term relationship [9]. Consistent and correct condom use during high-risk sexual behaviors could depress the risk of HIV infection by approximately 69% [30,31]. Thus, the frequencies of condom use during these sexual acts in the past year and in the last sexual encounter were asked in order to evaluate condom use patterns (e.g., never, sometimes, and always). Consistent condom use was defined as “Always,” and options of “Never” and “Sometimes” were regarded as inconsistent condom use. The lifetime history of drug use was also enquired (i.e., the respondents answered with a “yes” or “no”). 

#### 2.3.2. Sero-Testing

Exactly 5 mL of venous blood sample was collected from each participant for HIV, syphilis, and HCV antibody testing by using the standard protocol and laboratory methods of the National Center for AIDS Prevention and Control in China [32]. For HIV and HCV, the samples that tested positive in the highly sensitive enzyme-linked immunosorbent assay (ELISA) test ELISA-1 were subjected to a highly specific ELISA (ELISA-2), and then were confirmed to be positive [33]. Syphilis antibodies were screened using the ELISA test, and positives were confirmed by treponema pallidum particle assay (TPPA) [34]. All the screening and confirmatory tests were conducted at designated and certified laboratories in local CDC or CDC-accredited hospitals. The anonymous test results were linked with the interview data through the UIDs. 

### 2.4. Data Quality Control

The surveys and the serological testing were conducted by doctors at the Center for Disease Control (CDC) in Chongqing, China. The same sampling design and the training module for doctors were used at all sites. Among the eligible subjects, those who had already participated in this survey in Chongqing during the same year were excluded. The answers to the questionnaire were checked by senior investigators to ensure the completeness of the survey. The results were entered into the database by designated personnel who were blind to the field survey.

### 2.5. Data Analysis

Collected questionnaire data, along with the results of the serological tests in our serial cross-sectional surveys, were entered into the database management system of the National HIV/AIDS surveillance in China and exported from the system. The respondent’s socio-demographic characteristics; infection rates of HIV, syphilis, and HCV; and HIV-related knowledge and behaviors were described by number and percentages. The Chi-square test and Binary Logistic Regression were conducted to observe the differences between different age groups. The trends from 2010 to 2018 were tested using the Cochran–Armitage trend test. All analyses were conducted with the SPSS20.0 software. 

## 3. Results

### 3.1. Socio-Demographic Characteristics 

Data were obtained from 11,252 MMWs. Among them, most (46.9% in 2010 to 44.3% in 2018) were aged 30–49 years; more than 50% (63.9% in 2010 to 54.1% in 2018) were married; most (53.6% in 2010 to 74.1% in 2018) were from Chongqing and the remainder were from other provinces; the minority (23.1% in 2010 to 7.0% in 2018) had primary school or less education; the majority (43.4% in 2011 to 51.2% in 2018) had lived in Chongqing for more than 1 year; and the minority (0.3% in 2011 to 0.0% in 2018) were miners (see Table 1).

### 3.2. Prevalence and Trends of HIV, Syphilis, and HCV 

During the study period, the overall HIV prevalence was 0.6% with an increasing trend (0.2% in 2010 to 0.9% in 2018, *p* < 0.001) among MMWs in Chongqing. Unlike HIV, the overall prevalence of HCV was 0.5% with a decreasing trend (0.5% in 2010 to 0.4% in 2018, *p* < 0.001). The prevalence of syphilis was 0.6% in 2010 and 0.7% in 2018 and did not significantly change among MMWs in Chongqing. The prevalence of HIV and HCV dropped considerably from 2014 to 2016. Syphilis prevalence also dropped greatly during 2014–2015 and 2016–2017. However, syphilis prevalence increased from 2015–2016 and HIV prevalence increased from 2016–2017 (Figure 1).

The overall prevalence of syphilis was 1.3% in the ≥50 age group, 1.0% in the 30–49 age group, and higher than 0.3% in the 16-29 group (X^2^ = 19.527, *p* < 0.001; Figure 2).

### 3.3. HIV-Related Knowledge and Behaviors

#### 3.3.1. HIV-Related Knowledge

Approximately 9136 (81.2%) participants had correct knowledge regarding HIV, which showed an uptrend (80.2% in 2010 to 80.6% in 2018, *p* < 0.001; see Table 2). The 16–29 ([Odds Ratio] OR: 1.575; 95% CI [Confidence Interval]: 1.380–1.798; *p* < 0.001) and 30–49 (OR: 1.697; 95% CI: 1.495–1.926; *p* < 0.001) age groups had 1.575 and 1.697 times correct HIV-related knowledge more than the ≥50 age group, respectively (Table 3).

#### 3.3.2. HIV-Related Risk Behaviors

Approximately 1346 (14.3%) participants had engaged in commercial sex with a female in the past year with an increasing trend (7.7% in 2010 to 13.3% in 2018, *p* < 0.001). Approximately 1245 (12.9%) and 40 (0.4%) participants engaged in non-regular sex with a female and anal sex with a male in the past year, while significant trends were not found among MMWs in Chongqing from 2010 to 2018. In addition, 25 (0.2%) respondents had experience with drug usage, and no change was observed from 2010 to 2018 (Table 2).

The risk of engaging in commercial sex with a female in the past year in the 16–29 age group was 0.768 times (OR: 0.768; 95% CI: 0.643–0.917; *p* = 0.003) less than that in the ≥50 age group. The risk of engaging in non-regular sex with a female in the past year in the 16–29 (OR: 2.819; 95% CI: 2.317–3.431; *p* < 0.001) and 30–49 (OR: 1.432; 95% CI: 1.184–1.733: *p* < 0.001) age groups were 2.819 and 1.432 times more than that in the ≥50 age group, respectively. The risk of engaging in anal sex with a male in the past year in the 16–29 age group was 6.333 times (OR: 6.333; 95% CI: 1.468–27.327; *p* < 0.013) more than that in the ≥50 age; group (Table 3).

#### 3.3.3. Condom Use During HIV-Related Risk Behaviors

The proportion of consistent condom use during commercial sex with a female in the past year (20.5% in 2010 to 54.0% in 2018, *p* < 0.001) and condom use in the last commercial sex (48.6% in 2010 to 72.1% in 2018, *p* < 0.020) dramatically increased. The proportion of consistent condom use during non-regular sex with a female in the past year (10.9% in 2010 to 47.3% in 2018, *p* < 0.001) and condom use in the last non-regular sex (40.8% in 2010 to 71.1% in 2018, *p* < 0.001) increased remarkably. Approximately 9 (22.5%) participants used condoms consistently during anal sex with a male in the past year and 18 (45.0%) participants used condoms in the last anal sex, but no significant trends were observed among MMWs in Chongqing from 2010 to 2018 (Table 2).

The possibilities of consistent condom use during commercial sex with a female in the past year in the 16–29 (OR: 2.606; 95% CI: 1.847–3.677; *p* < 0.001) and 30–49 (OR: 1.632; 95% CI: 1.214–2.195; *p* = 0.001) age groups were 2.606 and 1.632 times more than that in the ≥50 age group, respectively. The possibilities of condom use in the last commercial sex in the 16–29 (OR: 1.805; 95% CI: 1.258–2.589; *p* = 0.001) and 30–49 (OR: 1.360; 95% CI: 1.016–1.821; *p* = 0.039) age groups were 1.805 and 1.360 times more than that in the ≥50 age group, respectively. The possibilities of consistent condom use during non-regular sex with a female in the past year (OR: 1.628; 95% CI: 1.066–2.484; *p* = 0.024) and condom use in the last non-regular sex (OR: 1.671; 95% CI: 1.148–2.433; *p* = 0.007) in the 16–29 age group were 1.628 and 1.671 times more than those in the ≥50 age group, respectively (Table 3).

## 4. Discussion

Knowing the trends of infectious diseases such as HIV, syphilis, and HCV is critical for their control in China, as they cumulatively constitute a worrisome public health challenge; particularly among high-risk groups including MMWs. Chongqing is one of the busiest regions for the inflow of China’s migrant population. Therefore, we investigated the prevalence of these diseases and relevant knowledge and behaviors of MMWs in Chongqing.

The results of the survey revealed that over half (56.3%) of the respondents only completed junior high school or less. Thus, MMWs tended to be poorly educated and consequently engaged in behaviors that placed them at risk of acquiring HIV, as has been found in other studies of internal Chinese migrants [10,35].

In general, the HIV prevalence for 2010–2018 in our study was 0.6%, which was approximately 11 times higher than the overall national HIV prevalence of China in 2011 (0.058%). This result illustrated that MMWs in Chongqing were, indeed, at a high risk of HIV infection. Sentinel surveillance results for nine consecutive years showed that the positive rate of HIV among MMWs in Chongqing was on the rise (0.2% to 0.9%, *p* < 0.001). In the past few years, China has dramatically increased its coverage of screening and treatment for HIV, particularly among high-risk populations [33]. Although new occurrences have probably decreased slightly, due to some behavioral changes such as increasing consistent condom use during commercial sex (20.5% to 54.0%, *p* < 0.001) and non-regular sex (10.9% to 47.3%, *p* < 0.001) with females in the past year, survival has improved owing to the increased coverage of antiretroviral therapy. Hence, an overall increase in the prevalence of HIV may occur. Generally, the infection rate of HCV among MMWs in Chongqing was 0.5%, which was in agreement with the prevalence (0.5%) estimated by the national surveillance system between 2009 and 2012 [36]. HCV prevalence showed a downward trend, from 0.5% in 2010 to 0.4% in 2018, probably as the risk of HCV transmission is mainly from unsafe blood injections, and such risks have been reduced substantially since the introduction of nucleic acid testing in blood banks by the Chinese government [37]. The average prevalence of syphilis was 0.8% higher than that in eastern China (0.48%), probably due to the differences in educational level [16]. 

Interestingly, we found that the prevalence of HIV and HCV both dropped considerably from 2014 to 2016; further, syphilis prevalence also decreased greatly during 2014–2015 and 2016–2017. This may be due to the establishment of the trinity AIDS prevention and control model, including monitoring, treatment, and follow-up in Chongqing from 2013 to 2014 [38]. It is likely that the increased HIV prevalence from 2016–2017 is a result of all AIDS patients being recommended antiretroviral therapy since 2016, leading to more survival rates. 

In recent years, a few studies have found that the problem of middle-aged and elderly people infected with HIV has become increasingly prominent [39]. Interestingly, we found that the elderly (≥50 years old) and middle-aged (30–49 years old) MMWs were more likely to suffer from syphilis than the young group (16–29 years old). As people in China mainly consider sexual behaviors to be primary activities of young adults, less attention has been paid to elderly men infected with STIs participating in high-risk sexual behaviors [40]. However, we observed that the 16–29 age group was less likely to engage in commercial sex in the past year than the ≥50 age group. The possibilities of consistent condom use during commercial sex in the past year in the 16–29 and 30–49 age groups were 2.606 and 1.632 times more than that in the ≥50 age group; the possibilities of condom use in the last commercial sex in the 16–29 and 30–49 age groups were 1.805 and 1.360 times more than that in the ≥50 age group; and the possibilities of consistent condom use during non-regular sex in the past year and condom use in the last non-regular sex in the 16–29 age group were 1.628 and 1.671 times more than those in the ≥50 age group. The lesser frequency of using condoms during high-risk behaviors may have led to the higher STI prevalence in the ≥50 age group. Another potential reason is that STIs can easily be transmitted among elderly men, as old age can cause a decline in cellular and humoral immune function. This condition could also be explained by the fact that young people are better educated and have more knowledge of HIV than their elderly counterparts, as supported by our results [9]. 

Given poor HIV-related awareness, persistent high-risk sexual behaviors could more likely result in HIV [41]. The surveillance data indicated that the correct HIV awareness rates among MMWs in Chongqing were less than 80% in 2011, 2013, and 2014; which was far from enough. Although some MMWs had a high awareness of HIV transmission, they were short of comprehensive HIV knowledge, such as prevention knowledge and testing services. A significant increase in the engagement in commercial sex (7.7% to 13.3%, *p* < 0.001) was observed among the participants during the study period, probably due to the increasing use of the internet to find high-risk sexual partners, such as sex workers [42]. This appeared to be a positive sign, considering the role of such behaviors in increasing the risk of acquiring HIV and other STIs [43]. We also observed a relatively low percentage of consistent condom use during commercial (40.3%) and non-regular sex (31.1%), similar to the findings of other studies [44,45,46]. Unfortunately, we did not investigate vaginal, oral, and anal sexualities during commercial and non-regular sexes separately. A large number of studies indicated that unprotected commercial sexual behavior is one of the main routes of HIV/AIDS transmission [47,48]. Several Chinese studies have explained that the primary reason given for condom use among migrants is contraception, instead of disease prevention [4]. 

Most rural-to-urban MMWs work in the city for a short period and then return to their home villages. Migrant workers have, thus, been identified as a “bridge population” for HIV infection from the high-risk population to the general population [49]. The primary direction of such spread has long been assumed to originate from migrants who become infected while away from home and then infect their rural partners when they return, which contributes to more rapid transmission of HIV [50]. HIV prevention for this enormous and increasing population would not only protect themselves, but also reduce the spread of HIV to their partners, family members, and their communities [8]. 

A long and continuous observation time and protocol uniformity were the main strengths of our study. Our work also had several important limitations, however. First, non-participation could be affected by their HIV sero-status, and subjects living with HIV could be selected repeatedly in nine years, which can have a potential impact on the overall HIV epidemic. Second, some recall bias may exist in some related behaviors in the previous year or in the last sexual encounter. Third, in these interviews, self-reported information bias due to the social desirability was another potential vulnerability, as respondents may have been influenced by what they feel is “correct.” Fourth, the MMWs in our study were from only four occupational clusters of three districts in Chongqing, which limits the generalizability of the findings. Fifth, the cross-sectional design of our questionnaire may ignore other mixed factors. Finally, permanent MMWs were not included, but they may suffer from adverse conditions similar to non-permanent MMWs. Despite these limitations, this study has made important findings about the current and possible future contributions of MMWs to Chongqing’s STI epidemic.

## 5. Conclusions

The overall HIV prevalence has slightly increased over the past nine years among MMWs in Chongqing, and HCV prevalence has decreased simultaneously. Compared with the young group, middle-aged and elderly MMWs were more likely to be infected with syphilis. Elderly MMWs were more inclined to engage in unprotected high-risk behaviors. Thus, evidence-based prevention and appropriate education related to increasing commercial sex and inadequate condom use during high-risk behaviors should be expanded for this vulnerable population, especially those aged 50 years and above. 

Our study provides a preliminary investigation into the HIV, syphilis, and HCV epidemics among MMWs in Chongqing. Future work could adopt similar techniques to understand the epidemics in other Chinese cities or provinces, in order to help guide public health policies toward a directed STI intervention in China. 

## Figures and Tables

**Figure 1 ijerph-17-00875-f001:**
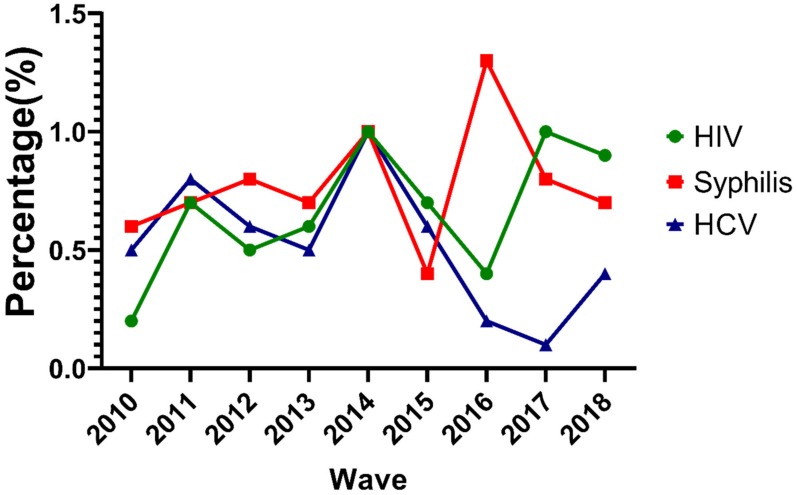
Trends of HIV, syphilis, and hepatitis C virus (HCV) prevalence for male migrant workers (MMWs) in Chongqing from 2010 to 2018.

**Figure 2 ijerph-17-00875-f002:**
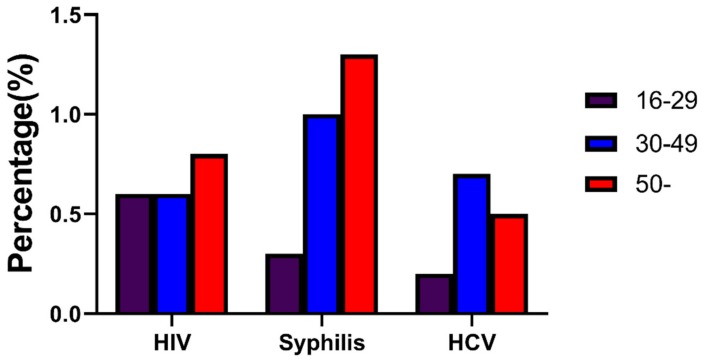
Overall prevalence of HIV, syphilis, and HCV for MMWs of three age groups in Chongqing from 2010 to 2018.

**Table 1 ijerph-17-00875-t001:** Socio-demographic characteristics of male migrant workers (MMWs) between 2010 and 2018 in Chongqing (*N* = 11,252).

Variables	2010(%)	2011(%)	2012(%)	2013(%)	2014(%)	2015(%)	2016(%)	2017(%)	2018(%)
	(*N* = 1208)	(*N* = 1203)	(*N* = 1200)	(*N* = 1600)	(*N* = 1203)	(*N* = 1200)	(*N* = 1200)	(*N* = 1227)	(*N* = 1211)
**Age (year)**									
16–29	(465)38.5%	(459)38.2%	(605)50.4%	(584)36.5%	(352)29.3%	(397)33.1%	(317)26.4%	(346)28.2%	(473)39.1%
30–49	(566)46.9%	(603)50.1%	(461)38.4%	(827)51.7%	(570)47.4%	(554)46.2%	(670)55.8%	(657)53.5%	(536)44.3%
50–	(177)14.7%	(141)11.7%	(134)11.2%	(189)11.8%	(281)23.4%	(249)20.8%	(213)17.8%	(224)18.3%	(202)16.7%
**Marital status**									
Never married	(311)25.7%	(341)28.3%	(380)31.7%	(332)20.8%	(188)15.6%	(179)14.9%	(170)14.2%	(153)12.5%	(349)28.8%
Married	(772)63.9%	(748)62.2%	(609)50.8%	(1126)70.4%	(841)69.9%	(818)68.2%	(832)69.3%	(776)63.2%	(655)54.1%
Never married but cohabited	(111)9.2%	(94)7.8%	(200)16.7%	(124)7.8%	(122)10.1%	(181)15.1%	(176)14.7%	(253)20.6%	(165)13.6%
Divorced or widowed	(14)1.2%	(20)1.7%	(11)0.9%	(18)1.1%	(52)4.3%	(22)1.8%	(22)1.8%	(45)3.7%	(42)3.5%
**Household register**									
Chongqing	(648)53.6%	(893)74.2%	(868)72.3%	(632)39.5%	(577)48.0%	(922)76.8%	(879)73.2%	(978)79.7%	(897)74.1%
Other province	(560)46.4%	(310)25.8%	(332)27.7%	(968)60.5%	(626)52.0%	(278)23.2%	(321)26.8%	(249)20.3%	(314)25.9%
Education									
Primary or less	(278)23.1%	(225)18.7%	(146)12.2%	(309)19.3%	(227)18.9%	(107)8.9%	(132)11.0%	(101)8.3%	(85)7.0%
Junior high school	(639)53.1%	(508)42.3%	(453)37.8%	(684)42.8%	(506)42.1%	(475)39.6%	(551)45.9%	(467)38.2%	(431)35.7%
Senior high school or above	(286)23.8%	(469)39.0%	(601)50.1%	(607)37.9%	(470)39.1%	(618)51.5%	(517)43.1%	(655)53.6%	(692)57.3%
**Local residence time**									
<3 months	(135)11.2%	(99)8.2%	(114)9.5%	(72)4.5%	(142)11.8%	(86)7.2%	(106)8.8%	(134)11.0%	(209)17.3%
3–6 months	(278)23.1%	(142)11.8%	(256)21.3%	(126)7.9%	(161)13.4%	(122)10.2%	(140)11.7%	(202)16.5%	(156)12.9%
6–12 months	(268)22.3%	(172)14.3%	(346)28.8%	(281)17.6%	(239)19.9%	(259)21.6%	(319)26.6%	(334)27.3%	(225)18.6%
>1 year	(523)43.4%	(790)65.7%	(484)40.3%	(1121)70.1%	(661)54.9%	(733)61.1%	(635)52.9%	(553)45.2%	(618)51.2%
**Sample source**									
Construction worker	(572)47.9%	(215)17.9%	(220)18.3%	(563)35.2%	(379)31.5%	(214)17.8%	(291)24.2%	(272)22.1%	(193)15.9%
Miner	(3)0.3%	(0)0.0%	(1)0.1%	(16)1.0%	(6)0.5%	(9)0.8%	(8)0.7%	(18)1.5%	(0)0.0%
Enterprise worker	(226)18.9%	(232)19.3%	(359)29.9%	(286)17.9%	(425)35.3%	(444)37.0%	(444)36.9%	(424)34.5%	(372)30.7%
Business operation	(166)13.9%	(699)58.1%	(547)45.6%	(408)25.5%	(152)12.6%	(205)17.1%	(204)16.9%	(116)9.4%	(165)13.6%
Others	(226)18.9%	(57)4.7%	(73)6.1%	(327)20.4%	(241)20.0%	(328)27.3%	(257)21.3%	(398)32.4%	(482)39.8%

**Table 2 ijerph-17-00875-t002:** HIV-related knowledge and behaviors among MMWs between 2010 and 2018 in Chongqing.

Variables	2010(%)	2011(%)	2012(%)	2013(%)	2014(%)	2015(%)	2016(%)	2017(%)	2018(%)	Chi-square Value	*p*-Value for Trend
Correct HIV-related knowledge	(969/1208)	(903/1203)	(996/1200)	(1237/1600)	(923/1203)	(1080/1200)	(1029/1200)	(1023/1227)	(976/1211)	27.503	<0.001
	80.2%	75.1%	83.0%	77.3%	76.7%	90.0%	85.8%	83.4%	80.6%		
Engaged in commercial sex with female in the past year	(74/962)	(94/935)	(114/892)	(125/1172)	(176/1120)	(190/1074)	(298/1102)	(136/1132)	(139/1046)	55.227	<0.001
	7.7%	10.1%	12.8%	10.7%	15.7%	17.7%	27.0%	12.0%	13.3%		
Consistently used condom during commercial sex with female in the past year	(15/73)	(36/94)	(49/114)	(43/125)	(70/177)	(73/194)	(146/298)	(63/136)	(75/139)	22.691	<0.001
	20.5%	38.3%	43.0%	34.4%	39.5%	37.6%	49.0%	46.3%	54.0%		
Used condom in last commercial sex with female	(36/74)	(68/94)	(87/113)	(82/125)	(128/175)	(117/194)	(217/298)	(105/135)	(98/136)	5.406	0.020
	48.6%	72.3%	77.0%	65.6%	73.1%	60.3%	72.8%	77.8%	72.1%		
Engaged in non-regular (unpaid) sex with female in the past year	(102/960)	(86/929)	(162/890)	(120/1387)	(180/1119)	(178/1084)	(173/1102)	(115/1132)	(129/1044)	2.757	0.097
	10.6%	9.3%	18.2%	8.7%	16.1%	16.4%	15.7%	10.2%	12.4%		
Consistently used condom during non-regular (unpaid) sex with female in the past year	(11/101)	(30/86)	(39/162)	(24/120)	(50/181)	(44/179)	(55/173)	(68/115)	(62/131)	52.519	<0.001
	10.9%	34.9%	24.1%	20.0%	27.6%	24.6%	31.8%	59.1%	47.3%		
Used condom in the last non-regular (unpaid) sex with female	(42/103)	(56/87)	(100/162)	(80/120)	(107/180)	(84/179)	(119/173)	(84/111)	(91/128)	16.579	<0.001
	40.8%	64.4%	61.7%	66.7%	59.4%	46.9%	68.8%	75.7%	71.1%		
Engaged in anal sex with male in the past year	(2/967)	(6/937)	(1/898)	(8/1174)	(6/1133)	(5/1082)	(7/1102)	(4/1128)	(1/1044)	0.143	0.705
	0.2%	0.6%	0.1%	0.7%	0.5%	0.5%	0.6%	0.4%	0.1%		
Consistently used condom during anal sex with male in the past year	(0/0)	(1/6)	(0/0)	(1/8)	(1/6)	(2/6)	(1/7)	(2/4)	(1/1)	2.763	0.096
	0.0%	16.7%	0.0%	12.5%	16.7%	33.3%	14.3%	50.0%	100.0%		
Used condom in the last anal sex with male	(0/0)	(3/6)	(0/0)	(2/8)	(3/5)	(3/6)	(4/7)	(2/4)	(1/1)	2.141	0.143
	0.0%	50.0%	0.0%	25.0%	60.0%	50.0%	57.1%	50.0%	100.0%		
Used drug in lifetime	(0/0)	(4/1203)	(3/1196)	(1/1597)	(3/1200)	(6/1197)	(4/1200)	(3/1227)	(1/1211)	0.459	0.498
	0.0%	0.3%	0.3%	0.1%	0.3%	0.5%	0.3%	0.2%	0.1%		

**Table 3 ijerph-17-00875-t003:** The distinctions of HIV-related knowledge and behaviors between different age groups.

Variable	β Value	Wald Value	*p*-Value	[Odds Ratio]OR (95%CI[Confidence Interval])
**Correct HIV-related knowledge** (reference group: ≥50)				
16–29	0.454	45.125	<0.001	1.575 (1.380–1.798)
30–49	0.529	66.736	<0.001	1.697 (1.495–1.926)
**Engaged in commercial sex with female in the past year** (reference group: ≥50)				
16–29	−0.264	8.537	0.003	0.768 (0.643–0.917)
30–49	−0.002	0.001	0.979	0.998 (0.858–1.161)
Consistently used condom during commercial sex with female in the past year (reference group: ≥50)				
16–29	0.958	29.754	<0.001	2.606 (1.847–3.677)
30–49	0.490	10.511	0.001	1.632 (1.214–2.195)
**Used condom in the last commercial sex with female** (reference group: ≥50)				
16–29	0.591	10.281	0.001	1.805 (1.258–2.589)
30–49	0.308	4.270	0.039	1.360 (1.016–1.821)
**Engaged in non-regular (unpaid) sex with female in the past year** (reference group: ≥50)				
16–29	1.036	107.086	<0.001	2.819 (2.317–3.431)
30–49	0.359	13.678	<0.001	1.432 (1.184–1.733)
**Consistently used condom during non-regular (unpaid) sex with female in the past year** (reference group: ≥50)				
16–29	0.487	5.099	0.024	1.628 (1.066–2.484)
30–49	0.249	1.350	0.245	1.282 (0.843–1.951)
**Used condom in the last non-regular (unpaid) sex with female** (reference group: ≥50)				
16–29	0.513	7.173	0.007	1.671 (1.148–2.433)
30–49	0.167	0.803	0.370	1.182 (0.820–1.703)
**Engaged in anal sex with male in the past year** (reference group: ≥50)				
16–29	1.846	6.121	0.013	6.333 (1.468–27.327)
30–49	1.230	2.749	0.097	3.423 (0.799–14.657)

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
