# Peer review of "Changing Trends of HIV, Syphilis, and Hepatitis C among Male Migrant Workers in Chongqing, China: Nine Consecutive Cross-Sectional Surveys, 2010–2018"

_ijerph, 2020, doi:10.3390/ijerph17030875_

Round 1

Reviewer 1 Report

Thank you for submitting the manuscript “Changing trends of HIV, syphilis, and hepatitis C among male migrant workers in Chongqing, China: Nine consecutive cross-sectional surveys, 2010–2018” to the International Journal of Environmental Research and Public Health. The manuscript is important in providing an examination of the public health interventions needed for the male migrate workers in China. The sample size is very impressive, taking into account the sensitive topic of the study and the vulnerable population it involves.

Please see my following comments on the manuscript:

Abstract

Please add “Hepatitis C” (HCV) when mentioning HCV in the first time.

Please add “sexually transmitted infection” (STI) before STI in the first time.

Introduction

Line 71: Please change the word “gainful” to “gaining”.

Line 72: Please add Male migrate workers before MMWs. 

Line 89: What is the full term of FSW? Please write in length before FSWs.

Line 95: Add the word “which” after the sentence “low educational levels”.

Please include a paragraph on sexual behavior and risk behaviors among MMWs. Please provide a definition to risk behaviors.

Methods

2.1 Design.

Please add if the study was ethically approved by an authorized entity/academic institution.

2.2. Recruitment

Line 119 – Please change the word “overwhelmingly” to “mainly” and add the percentage of male migrate after the word mainly.

Please include if incentives were provided to participants.

2.3. Socio-demographics… please change the title to “measures”.

Please add a sentence or two on how many scales and their names that were used in the study before elaborating on each scale. Reliability of scales are needed. If they were created for this study, please include such information. Also include a sentence that the study method included a survey and a serological testing, then elaborate on them and the way they were conducted.

Lines 148-149 – Please remove this sentence to be included in the literature review.

Line 158-159 – please remove this sentence to be included in the design section.

2.5 Data quality control

Line 173 - Trained staff – please provide more information on the background of the staff who conducted the surveys and the ones who took the serological testing.

Line 173 – The same sampling….followed” followed by what? This sentence is unclear.

It is not clear if the participants filled in the surveys and provided serological tests in different times during the years of the study, or if the participants only provided one survey and serological test during the study. When the authors talk about an increase or decrease in STDs, they want to be clear about the tested sample of participants. Otherwise it is a huge limitation of their study and needs to be included in the limitation section. 

2.6 Data analysis

The authors need to state in a clear way if their study is cross sectional or a longitudinal one!

Results

3.1. Socio-demographic characteristics. The wording provided is very limited and confusing. Please only state that the MMWs socio-demographic characteristics are presented in Table 1.

Line 228 – Please change the wording to “had experienced drugs usage”.

Lines 231-232 – was 0.768 (more or less?) times than… please clarify. The same comment for lines 234-235 and lines 237, 240, 243, 244.

Please edit the result section into more clear description by dividing the findings according to the scales used: risk behavior, condom usage…etc.

Please provide few sentence on figure 1. There was a huge drop in HIV in 2016, but an increase in 2017. This did not happen with HCV. Syphilis has also increased in 2016 and then dropped in 2017. If you present a figure, you need to explain a little bit about it, or at least mention the most important finding it represents. Also, the aims of your study were to offer public health interventions for the benefit of MMWs. Therefore, it could be very interesting to examine if condoms distribution by clinics or authorities were conducted in certain years, or if an educational campaign was implemented among this population in certain years, and therefore partly explain the drop in STDs in certain years. You may also consider other factors!   

Discussion

The discussion could benefit from an analysis of the findings not only focused on age differences, but also the differences in the prevalence of the diseases in different years.

Lines 290-291 – It is unacceptable to address women’s sexuality in their relationships as a “sexual supply” and accuse them in making their partners seek sexual services to meet their sexual needs!!!! This statement in the discussion is extremely sexist and unacceptable in academic manuscripts. It is also not supported by any literature review!!! Is this the reason why older men seek sexual services? Their wives? Really? I can think of other alternative reasons based in the literature of medicine and psychology. DELETE THIS REASON.

Line 308 – Please change the wording to “similar to the findings of other studies”.

The discussion needs to include findings relates to STDs that are transmitted via oral sex and not only via intercourse (in the paragraph starting with line 298).

Lines 314-315 – please provide the percentage of such phenomena and rewrite the paragraph. It seems that such statement can be understood that only the MMWs are the ones responsible for the spreading of HIV.

Line 318 – Delete “former partners” since they return to them when they reach home.  

Please add the following limitations. First, the study included MMWs from only three districts in China. Second, the study included only workers from four occupational clusters. Third, permanent MMWs were not included, but these workers can expand the understanding of STDs among this vulnerable group of men (who might suffer from low income, low access to clinics and low utilization of services similar to non-permanent workers). Fourth, cross sectional study design, as mentioned above. While the sample size in comparatively large (11, 252), it is still limited to this certain group pf MMWs and therefore the generalization of the findings to other MMWs groups is also limited. 

Abbreviations should be mentioned in their length at first including the abbreviations as I have mentioned above, rather than including them at the end of the manuscript after the reader has finished reading the manuscript.  

Reviewer 2 Report

The purpose of this study is unclear. The authors need to verify the purpose of this study.  The descriptions of this study are broad and lose the focus. The readers are difficult to read.  The abstract and text need to be modified.   

1. I don’t know what is the purpose of this study?
2. Why they put the three diseases together in on paper?
3. They want to investigate the risk of HIV infection? or risk of blood transmitted diseases?
4. I don't know why they only explore the KAP of HIV, but not hepatitis C?
5. I don't know why they only mention the condom use, not other risk factors, e.g. repeated syringe use?
6. Therefore I suggest the author to clarify their study purpose and re-write the paper.

Round 2

Reviewer 2 Report

This manuscript has been revised by authors as reviewers' comments. This version has got much improvement compared with the last version.  

Author Response

Dear Editor,

  We really appreciate all of your constructive notes. All of you have helped us improve the quality of this manuscript significantly. Our manuscript has been edited by editors of IJERPH. Thank you very much!

Best regards,

Mengliang Ye

Jan 27, 2020

This manuscript is a resubmission of an earlier submission. The following is a list of the peer review reports and author responses from that submission.